# The Elaboration of Effective Coatings for Photonic Crystal Chips in Optical Biosensors

**DOI:** 10.3390/polym14010152

**Published:** 2021-12-31

**Authors:** Svetlana Sizova, Ruslan Shakurov, Tatiana Mitko, Fedor Shirshikov, Daria Solovyeva, Valery Konopsky, Elena Alieva, Dmitry Klinov, Julia Bespyatykh, Dmitry Basmanov

**Affiliations:** 1Federal Research and Clinical Center of Physical-Chemical Medicine of Federal Medical Biological Agency, 1A Malaya Pirogovskaya St., 119435 Moscow, Russia; ruslan.shakurov@rcpcm.org (R.S.); tanya-mitko@yandex.ru (T.M.); shirshikov@rcpcm.org (F.S.); klinov.dmitry@mail.ru (D.K.); juliabespyatykh@gmail.com (J.B.); basmanov.dmitry@gmail.com (D.B.); 2Department of Biomaterials and Bionanotechnology, Shemyakin & Ovchinnikov Institute of Bioorganic Chemistry RAS, 16/10 Miklukho-Maklaya St., 117997 Moscow, Russia; d.solovieva@mail.ru; 3Department of Molecular and Translational Medicine, Moscow Institute of Physics and Technology, 9 Institutskiy Per., 141701 Dolgoprudny, Russia; 4Expertise Department in Anti-Doping and Drug Control, Mendeleev University of Chemical Technology of Russia, 9, Miusskaya Sq., 125047 Moscow, Russia; 5Solid State Spectroscopy Department, Institute of Spectroscopy RAS, 5 Fizicheskaya St., 108840 Moscow, Russia; konopsky@gmail.com (V.K.); alieva@isan.troitsk.ru (E.A.)

**Keywords:** photonic crystal surface mode biosensor, biosensor coating, dextran, microfluidics, optical surface waves, protein binding, label-free detection, biomolecular interaction

## Abstract

Here, we propose and study several types of quartz surface coatings designed for the high-performance sorption of biomolecules and their subsequent detection by a photonic crystal surface mode (PC SM) biosensor. The deposition and sorption of biomolecules are revealed by analyzing changes in the propagation parameters of optical modes on the surface of a photonic crystal (PC). The method makes it possible to measure molecular and cellular affinity interactions in real time by independently recording the values of the angle of total internal reflection and the angle of excitation of the surface wave on the surface of the PC. A series of dextrans with various anchor groups (aldehyde, carboxy, epoxy) suitable for binding with bioligands have been studied. We have carried out comparative experiments with dextrans with other molecular weights. The results confirmed that dextran with a Mw of 500 kDa and anchor epoxy groups have a promising potential as a matrix for the detection of proteins in optical biosensors. The proposed approach would make it possible to enhance the sensitivity of the PC SM biosensor and also permit studying the binding process of low molecular weight molecules in real time.

## 1. Introduction

Strategies for attaching biomolecules on a sensitive surface of biosensor substrate while maintaining its identity, structural conformation, and functionality are an evolving field of research. This can lead to the advances in the development of rapid and highly sensitive biosensors. Surface modification plays a significant role in detecting and characterizing molecular and cell affinal interactions on the biosensor surface. Involved chemical surface preparation is generally required to induce surface functional groups for protein immobilization. Among the different approaches, the most common methods to carry out such functionalization are electro- and photopolymerization [1,2], self-assembled monolayer [3,4], plasma polymerization [5,6], layer-by-layer deposition [7] and liquid phase deposition [8]. Such modifications allow the control of surface properties and the addition of new functionalities such as biocompatibility, chemical stability, antimicrobial activity, new electrical and optical properties, etc. Nevertheless, these methods may be complex, limited to certain surfaces, laborious, time consuming, and sometimes poorly reproducible.

To date, various methods have been developed for preparing and recharging the surface of a biosensor substrate [9,10,11]. For example, the methods used in the surface plasmon resonance (SPR) technique may, theoretically, also be suitable for modifying the surface of a photonic crystal (PC). The SPR technique is based on the excitation of the surface plasmon polaritons along a metal/dielectric interface by incident light with a wave vector matching that of the SPR. This label-free method is sensitive, rapid, and does not require complex preliminary sample preparation. In the SPR approach, a thin gold film on the solid support is coated with marker-specific antibodies, and an increase in the adlayer mass due to the association of soluble antigen with marker-specific antibodies on the surface of the film is recorded as a change in the refractive index [12]. Film surface modification with alkanethiol molecules (aminopropyltriethoxysilane, glycidylpropyltrimethoxysilane, etc.) allows the attachment of an immobilization matrix [13]. The immobilization matrix minimizes nonspecific binding of the ligand to the adhesive linker layer. It can be a layer of 11-mercaptoundecanoic acid with carboxyl functional group on the surface [14] or hydrophilic polymers, such as chitosan [15], acrylic acid [16], carboxymethyl dextran [17], etc.

One more label-free method is a high-precision multi-parameter biomarkers detection using the photonic crystal surface mode (PC SM) method. PC is a material characterized by the periodic modulation of refractive indices to the scale of the light wavelength. Such structures support long-range propagation surface optical waves along its outer surface. These waves, hereafter PC SM, which are also called “photonic band-gap surface modes”, “modes of (asymmetric) planar Bragg waveguide”, “surface waves in periodic layered medium”, ”photonic crystal surface waves”, ”optical Bloch surface waves”, and ”surface waves in multilayer coating” were studied in the 1970s, both theoretically [18,19] and experimentally [20]. Such 1D PC structures make it possible to detect both types of polarization (parallel and perpendicular) of the light wave and separately record the thickness of the surface adlayer and the refractive index of the liquid phase, so that the result of changing the parameters of the reflected signal carries information about the fact of interaction of the soluble analyte with the functionalized surface of the PC. It is known that the PC SM method is distinguished by higher sensitivity, also reduced analysis time and material consumption, in comparison with the SPR method [21]. The possibility of long-term exposure of the PC in a flow-through system without reducing its functional activity significantly increases the economic attractiveness of its use.

Surface modification plays a pivotal role for biomolecules detection and the characterization of diverse interactions on the sensor chip surface [22,23]. There has been recent interest in polysaccharide coatings for biomaterials applications and for biosensors. Polysaccharides have been used both to enhance and/or activate biointeraction as well as to increase the antiadhesive properties of biomaterials [24,25,26]. The performance of such coatings may depend markedly on factors such as the molecular weight, degree of branching and mode of fabrication [27]. Carboxymethyldextran (CMD) with Mw 500 kDa as a matrix for the covalent immobilization of ligands was first used by BIAcore [17] and successfully applied by SPR technology until now. CMD was used for the fabrication of a 3D Quartz crystal microbalance (QCM) biosensor for the real-time detection of biomolecular interactions [28]. The spin-coating deposition of dextran was implemented in [29] to increase the binding capacity of the surface of 1D PC. However, studies devoted to the modification of sensitive surface of PC SM biosensors using dextrans of different molecular weights and different active groups have not been carried out. In this study, we propose an approach to increase the sensitivity of the PC SM biosensor based on preliminary treating the PC chip sensitive surface with organosilanes followed by the formation of a 3D-branched coating from functionalized dextran. The sorption capacity of the dextran functionalized coatings and detection capacity of the PC SM biosensor were demonstrated using bovin serum albumin (BSA) as a model protein and low molecular weight biomolecules such as oligonucleotides and compared with sorption capacity and detection capacity of a control—planar the PC chip surface. The results confirmed that dextran with a molecular weight of 500 kDa and anchor epoxy groups have a promising potential as a matrix of optical biosensors for the detection of proteins and biomolecules of low molecular weight. We suppose that the proposed approach would make it possible to enhance the sensitivity of the PC SM biosensor and permit studying the low molecular weight molecules’ binding process in real time.

## 2. Materials and Methods

### 2.1. Materials

Dextran T500 (Mw ≈500 kDa) was obtained from Pharmacia (Uppsala, Sweden). Dextran with Mw 5 kDa and 50 kDa (analytical standard for GPC), phosphate-buffered saline (PBS), (3-aminopropyl)triethoxysilane (APTES, 99%), glutaraldehyde (GA), polyallylamine (PAA), epichlorohydrin (ECH, 99%), sodium periodate, bovine serum albumin (BSA, fatty acid free, low endotoxin, lyophilized powder, BioReagent, suitable for cell culture, ≥96%), and diethylene glycol dimethyl ether (diglyme, anhydrous, 99.5%) were obtained from Sigma-Aldrich (St. Louis, MO, USA). Carboxymethyldextran (“CM-dextran”, 500 kDa) was obtained from TdB Labs (Uppsala, Sweden). N-Hydroxysulfosuccinimide (sulfo-NHS) and 1-ethyl-3-(3-dimethylaminopropyl) carbodiimide (EDC) were purchased from Thermo Scientific (Waltham, MA, USA). Streptavidin (STP) was purchased from New England Biolabs (Ipswich, MA, USA). Other chemicals were of analytical grade and used without additional purification. All oligonucleotides (Table 1) were synthesized and purified by Lytech Co. Ltd. (Moscow, Russia) and solubilized in milli Q water. 3′-Biotin-TEG CPG from Glen Research was used for 3′-biotinylation of oligosensors.

APTES solution (5%), PAA solution (0.1 mg/mL) and GA solution (0.1%) were prepared with milli Q water (Millipore, Merck KGaA, Darmstadt, Germany) of highest purification with resistivity 18 MΩ cm. BSA was dissolved in PBS to obtain solutions with concentrations 0.1 mg/mL, 0.01 mg/mL, and 0.001 mg/mL. STP was dissolved in PBS to obtain solutions with concentration 0.005 mg/mL.

### 2.2. Photonic Crystal Surface Modes Detection System

EVA 2.0, a label-free PC SM biosensor with an independent recording of the critical angle of the liquid, was employed in this study (Figure 1) [21,30]. To obtain statistically reliable results, each experiment was repeated 3 times.

The desirable 1D PC structure was theoretically deduced using a previously described impedance approach [31]. The following 1D PC structure was designed by this method and used in the present experiments: (BK-7-substrate)/H (LH)^2^ L′/(water), where L is a SiO_2_ layer with thickness d_1_ = 231.3 nm, H is a TiO_2_ layer with d_2_ = 74.6 nm, and L′ is a SiO_2_ layer with d_3_ = 382.5 nm. The SiO_2_/TiO_2_ 6-layer structure (started from TiO_2_ and finished by SiO_2_ layers) was deposited by magnetron sputtering. The prism and the glass plate substrate are made from BK-7 glass. The RIs of the substrate, SiO_2_, TiO_2_ and water at λ = 659 nm, are n_0_ = 1.514, n_1_ = n_3_ = 1.48, n_2_ = 2.18, and n_e_ = 1.33, respectively.

### 2.3. Synthesis of Aldehyde Dextran

Aldehyde dextran (AD) was obtained via periodate oxidation of dextran with a molecular weight of 500 kDa according to an adapted protocol [32,33]. Briefly, 10 g of dextran was dissolved in 200 mL of distilled water, and sodium periodate was added in molar ratios of 1:10 (IO_4_/glucose units). The solution was stirred in the dark at room temperature for 2 h and after that was dialyzed against Milli-Q water using a Slide-A-Lyzer 3.5 K MWCO Dialysis Cassettes (Thermo Scientific, Rockford, IL, USA) for 48 h. The final products were dried in an incubator at 50 °C for 24 h. The number of aldehyde groups in AD was determined using p-nitrophenylhydrazine, which reacts equimolarly with an aldehyde group in the Appendix A. The method was adapted from [34].

### 2.4. Synthesis of Epoxydextran

Dextran with epoxy groups (ED) was obtained according to an adapted protocol [35]. Dextran with Mw 500 kDa (5 mg, 3.1 × 10^−2^ mol) was dissolved in 1 mL of a mixture of water/diglyme (1:1 ratio, *v*/*v*) in a reactor with constant stirring in an inert atmosphere until the dextran was completely dissolved. Then, 5 μL of an aqueous solution of NaOH (0.01 M) was added to the reactor, and the reaction mixture was intensively stirred for 30 min with heating to 50 °C. After the complete dissolution of all components of the reaction mixture, 30 μL (0.0384 mol) of ECH was added, and the mixture was stirred with cooling to 50 °C within 30 min. The product was purified by dialysis against Milli-Q water using a Slide-A-Lyzer 3.5 K MWCO Dialysis Cassettes (Thermo Scientific, Rockford, IL, USA) for 10 h.

### 2.5. Fabrication of the PC Biochip: Activation and Functionalization

The method involves: (a) the PC chip surface cleaning (detergent/water/ethanol ultrasonic cleaning, air plasma treatment); (b) generation of silane layer using organosilane reagent (APTES); and (c) immobilization of modified dextran with different anchor groups (carboxy-, aldehyde-, epoxy-) on the APTES-treated PC chip surface.

The PCs were cleaned in detergent Hellmanex (Hellma, Muellheim, Germany), thoroughly rinsed, and ultrasonicated with double-distilled water and ethanol three times and dried under nitrogen flow. Then, the clean PC chips were treated in the plasma cleaner Zepto W6 (13.56 MHz/100 W, Diener Electronic, Ebhausen, Germany) for 10 min at an air pressure 600 to 800 mbar. Plasma treatment controls the hydrophilicity of the surface (makes it more OH groups on the surface) to improve subsequent coating with functional groups [36]. After the cleaning step, the PC chips were immersed in APTES (5%, *v*/*v*) solution for 30 min, rinsed with double-distilled water, and baked for 30 min at 120 °C to remove the moisture (dehydration) present on the surface. The optimal APTES concentration and silanization time were previously selected [36,37]. The activated chip was placed into a flow cell; the inlet and outlet tubes were mounted. A peristaltic pump was used to ensure a constant flow through the cell at a flow rate of 50 μL/min.

We used solutions of dextrans in PBS with concentration of 5 mg/mL in all experiments. Carboxy groups of CMD preliminary were activated with EDC/sulfo-NHS according to an adapted protocol [38]. Briefly, CMD was dissolved in 0.5 M MES buffer (pH 6.0), and sulfo-NHS (25 μL, 0.5 M) and EDC (25 μL, 2 M) were added to the CMD solution. The mixture allowed stirring for 30 min in order to activate carboxyl groups of CMD. Modified dextrans (AD, activated CMD, ED) were run through the flow cell with a volumetric flow rate of 50 μL/min. The dextrans were deposited on the silanized PC surface and kept there for 10 to 30 min until the signal reached a plateau. Then, the flow cell was rinsed with PBS until the signal reached a plateau. Immobilization of the modified dextrans (AD, CMD with activated carboxy groups, ED) to the PC surface was monitored in situ by the PC SM biosensor.

### 2.6. Proteins and Low Molecular Weight Biomolecules Binding Detection

A model protein BSA in PBS was used as a test of binding detection with the dextran modified PC chip surface. All experiments with BSA were carried out in PBS (pH 7.2). To conjugate model protein BSA with the PC chip surface, BSA solution in PBS was run through the flow cell until signal stabilization; then, the chip surface was rinsed with PBS for 2 min. BSA binding with the modified dextrans (AD, CMD with activated carboxy groups, ED) was monitored in situ by the PC SM biosensor.

The PC SM biosensor detection capacity of low molecular weight biomolecules interaction was verified using oligonucleotides designed for *Mycobacterium tuberculosis* spacer oligonucleotide typing (spoligotyping) [39]. For spoligotyping, new oligonucleotides were constructed based on the 43 spacer sequences in direct repeat (DR) locus of *M. tuberculosis* (strain H37Rv; NCBI Accession NC_000962.3; locus 3119000:3124000). Thus, the S43 oligonucleotide is a model ssDNA target of the spacer; the Z43 oligonucleotide is a complementary sequence for the detection of S43. All oligonucleotides were solubilized in PBS at a concentration of 25 pmol/mL.

We used the PCs with an ED-modified sensitive surface (as described in Section 2.5). The binding of the oligosensors to the ED-modified PC surface was carried out via biotin–STP interaction. In all experiments with oligonucleotides, at the first stage, STP (0.005 mg/mL in PBS) was incubated with a biotinylated oligosensor for 10 min at RT for preparing a complex. Then, the complex was bound to the ED-modified PC chip surface. To block unbound epoxy groups on the ED-modified PC surface, BSA (0.1 mg/mL in PBS) was flowed over the flow cell for a few minutes, after which the oligotarget was run over.

The STP-biotinilated oligosensor, blocking agent BSA, and oligotarget were run through a flow cell of the biosensor, at a flow rate of 50 μL/min, until the signal stabilized by reaching a plateau; then, the system was thoroughly rinsed with PBS solution after each stage. As a negative control of oligonucleotide hybridization, two additional oligonucleotides with random sequence were designed: SR, as a model target of another spacer; and ZR, as a model oligosensor that should show a lack of hybridization with any spacer sequence.

### 2.7. Atomic Force Microscopy

The thickness of the fabricated 3D dextran PC chip surface was measured by atomic force microscope (AFM) SFC050LNTF (NT-MDT, Moscow, Russia), and high-resolution AFM cantilevers HA_HR (TipsNano, Tallinn, Estonia) were used. Measurements were carried out in the liquid phase to avoid deformation of the sample upon drying. At least 3 images were recorded from different areas of the scratched sample. AFM images were processed by filters Fit line X and Extract plane. Polymer films for AFM measurements were synthesized on pure square quartz substrates with an area of 25 mm^2^. The typical sample roughness was 2–3 nm in all samples.

## 3. Results

The approach based on the PC SM detection has been developed as a label-free high-precision biosensing technique. It allows real-time monitoring of molecular and cellular interactions using independent recording of the total internal reflection angle and the excitation angle of the PC surface wave [21]. An important advantage of the detection method using the PC substrate is the possibility of simple regeneration of the surface of the used substrates by plasma cleaning and subsequent activation and conjugation with a new recognition protein. This possibility makes it possible to use the same PC substrate an infinite number of times as well as prepare pre-activated PC substrates for specific experiment.

Earlier, a novel approach based on the simultaneous quantitative detection of several human serum tumor markers using the PC SM registration approach was presented [12]. Planar 1D PC chips for PC SM biosensors are very easy to manufacture using standard multilayer coating equipment. The simultaneous immobilization on planar surfaces yields limited protein density, while three-dimensional (3D) structures have been employed for higher and controlled protein capture capacity (Figure 2), resulting in improved immunoassay sensitivity [40].

Visualization of molecular structures at the PC surface was performed using the Avogadro software, version 1.2.0 [41] and the PyMOL Molecular Graphics System, version 2.4.1 (Schrödinger, LLC, New York, NY, USA) (Figure 3). Experimental structures of molecules were obtained from the Protein Data Bank as follows: BSA (PDB ID: 4F5S) and dextran monomers (PDB ID 5OCA).

In this work, dextran chemically modified to introduce active anchor groups for protein immobilization was chosen as the base of a 3D biochip matrix. The dextran activation process consists of the formation of free active groups within the dextran molecule. Another method of activation is to introduce active groups into the dextran molecule using a linker. These techniques are often combined [42,43]. AD was chosen because aldehyde groups can react with amine groups of proteins and the PC surface. In this work, active aldehyde groups were obtained by oxidation with sodium periodate. The use of the periodate anion for the oxidative cleavage of dextran was first reported by L. Malaprade as early as 1928 [44]. AD was synthesized according to the protocol adapted from [32,33]. CMD with Mw 500 kDa was used as a commercial product.

ED were synthesized according to the adapted protocol [35]. As known, the products of dextran crosslinking with epichlorohydrin, which are formed in an alkaline medium, possess high biocompatibility. The interaction of primary and secondary amines with glycidyl derivatives prior to gel formation allows the additional functionalization of dextran, which can lead to the formation of crosslinked polymers and non-crosslinked derivatives [35,45,46].

The PC surface was preliminarily treated with plasma and APTES was applied to introduce amino groups on the PC sensitive surface. As a result, APTES bond to activated silanol groups through ethoxy groups in the form of a monolayer with high physicochemical stability [47,48,49], while the amino group remained reactive and available for further modification. A schematic view of the PC surface modification with APTES and GAA–HA is shown in Figure 4.

ED was deposited on the APTES-treated PC surface for 20 min. AD and CMD with activated carboxy groups were run through the flow cell with a volumetric flow rate of v = 50 μL/min. Dextrans in PBS with a concentration of 5 mg/mL were used in all experiments. The flow cell was rinsed with PBS until the signal reached a plateau after each novel layer on the PC surface chip.

The reaction of dextran with ECH, carried out under the described conditions, replaces 2′-4′ glycidyl groups with epoxy ones; the scheme of such a reaction is shown in Figure 5. We assume that the epoxidized dextran obtained in this way binds to free amine groups on the PC surface.

The immobilization of dextrans with active anchor groups (CMD, AD, ED) demonstrated the change in the adlayer thickness, which was retained after rinsing with water, so we can assume that all the dextrans covalently bond with the amino groups of APTES-treated the PC surface chip.

### 3.1. Comparison of Model Protein Binding Capacity on Dextrans with Anchor Carboxy, Aldehyde, and Epoxy Groups

In order to assess the specific binding of model protein to the functionalized surface of the PC chip, we investigated the change in the thickness of the surface layer of a biosensor based on a PC at successive stages of preliminary processing of its surface and carrying out a model experiment.

We used BSA as a model protein to detect biomolecule binding with the modified dextran layer. Solutions of BSA were run through the flow cell at a rate of 50 μL/min. All experiments with BSA were carried out in PBS (pH 7.2). To bind BSA with the PC chip surface, a 0.1 mg/mL solution in PBS was run through the flow cell until signal stabilization; then, the surface was rinsed with PBS for 2 min. As a comparison, we used two known methods for modifying the PC chip surface with APTES and with PAA and GA [30,37,50].

As shown from the data presented (Figure 6), the maximum increase in sorption capacity was observed in case of the PC chip surface modified with ED. According to the results, modification of the PC surface using ED with a Mw of 500 kDa gives an increase of approximately 20% compared to the control values.

### 3.2. Investigation of Sorption Capacity of ED with Different Molecular Weight

In order to investigate the influence of Mw of dextran, we used dextrans with other molecular weight. Dextrans with Mw 5 kDa and 50 kDa were treated with ECH according to our protocol and compared with ED with Mw 500 kDa for the binding capacity of BSA (Figure 7). All experiments were performed at the same conditions. The results confirmed that the monolayer of BSA bonds with the PC chip surface in all experiments, but the binding capacity of the ED-modified PC surface decreased with the decreasing of molecular weight of ED, i.e., ED Mw 500 kDa > ED Mw 50 kDa > ED Mw 5 kDa and was 3.5, 2.2 and 1.7 nm, respectively (Figure 7).

### 3.3. Normalized Sensograms of BSA Binding to ED

Figure 8 depicts normalized sensograms (change in the adlayer thickness as a function of time) for the binding affinity interaction between BSA at various concentrations and epoxidized dextran (ED) with Mw 500 kDa. We used BSA solutions with concentrations of 0.001, 0.01, and 0.1 mg/mL.

We assumed that there are no steric difficulties for BSA to penetrate through the matrix fabricated according to our protocol, so it should not use BSA in concentrations lower than 0.1 mg/mL. The results obtained in the experiments allow us to evaluate the binding capacity of the modified PC chip surface and made it possible to maximize the amount of the bounded model protein BSA.

### 3.4. Detection of Low Molecular Weight Biomolecules Interaction by PC SM Biosensor

The detection capacity of the interaction of oligonucleotide sequences with low Mw were carried out under the conditions of the formation of the PC SW biosensor allowed maximizing the amount of the bound model protein BSA. At the first stage, a complex STP–biotinylated oligosensor was prepared and bound to the ED-modified PC surface. As a comparison, we used the PCs with surfaces modified with APTES and the PC chip modified with PAA and GA (Figure 9).

The data obtained showed an increase the sorption capacity of the ED-modified PC surface by almost two times compared to the control surfaces modified with APTES and PAA and GA.

Here, we see that ED modification of the PC surface by almost two times improves the detection capacity of the target low molecular weight molecules–oligonucleotide in comparison with the control surfaces modified with APTES and PAA and GA. Compared to the PC surfaces, modified with APTES and PAA–GA, i.e., planar surfaces, the branched ED surface demonstrates higher immobilization and binding capacity. We assumed that there are no steric difficulties for small molecules to penetrate through the ED matrix. Detailed numerical and percentage values are given in Table 2.

The absence of binding of a random oligosensor and random oligotarget, random oligosensor and specific oligotarget, specific oligosensor and random oligotarget was demonstrated (Figure 10). In addition, it can be concluded that there is no nonspecific binding of any oligotarget to the modified PC surface.

Thus, the possibility of solving an important problem [51,52] of low molecular weight oligonucleotides in picomole quantities detecting on the ED-modified PC chip by the PC SM method was demonstrated.

### 3.5. Measuring of the Thickness of Fabricated 3D Dextran Matrix on PC Chip

The thickness of the fabricated 3D dextran PC chip surface was measured by AFM. To avoid deformation of the sample upon drying, all measurements were carried out in the liquid phase. The sample preparation and scanning process are schematically shown in Figure 11.

Appendix A shows typical AFM images and cross-sections; thickness data were averaged. It should also be noted that the thickness data for modified PC chip substrates are resultant, and the contribution of APTES and ED with different molecular weight to these values are not specified. In agreement with the data given in [47], we can conclude that we obtained a monolayer of APTES with a thickness of 10 nm on the PC chip surface.

## 4. Discussion

Here, we propose and study several types of quartz surface coatings designed for the high-performance sorption of biomolecules and their subsequent detection by the PC SM biosensor. The deposition of biomolecules is revealed by analyzing changes in the propagation parameters of the surface modes on the PC surface. The method makes it possible to measure molecular and cellular affinity interactions in real time by independently recording the values of the angle of total internal reflection and the angle of excitation of the surface wave on the PC surface. The biosensor represents a modified PC surface with a functionalized 3D biosensor matrix. Dextran was chosen as a base of the 3D biochip matrix. We chemically modified dextrans to get active anchor groups and investigated the dextrans with carboxy, aldehyde, and epoxy groups as a 3D matrix for protein sorption capacity; we used BSA as a model protein. We have carried out comparative experiments with an APTES-modified PC surface and PAA-GA system and showed that the sorption capacity of ED with a Mw of 500 kDa is higher than 20% in comparison with the well-studied methods of surface modification with APTES and the PAA–GA system (Figure 6). We assumed that these results were achieved due to the branched 3D structure of dextran on the PC surface. We also investigated the ED sorption capacity with different molecular weight on the model protein BSA. The results confirmed that the monolayer of BSA bonds with the PC chip surface modified with ED (Mw 5, 50, and 500 kDa), and the binding is higher in case of the PC surface treated with ED with a Mw of 500 kDa (Figure 7). We have demonstrated that dextran with a Mw of 500 kDa and anchor epoxy groups had the maximal sorption capacity and showed a promising potential as a matrix for optical biosensors for the detection of proteins and biomolecules of low molecular weight. Comparing the data on the sorption capacity of the modified PC surface, the molecular weight of dextrans, and AFM data, it is difficult to estimate a direct proportional relationship between these values. Probably, a much more complex process is taking place and requires further study. The proposed method for biomolecules sorption and their subsequent detection by the PC SM biosensor demonstrated the detection of interaction of complementary oligonucleotides Z43 and S43. ED modification of the PC surface almost two times improved the detection capacity of oligotarget–oligonucleotide S43, compared to the control surfaces modified with APTES and PAA and GA. We believe that the branched 3D coating with active anchor groups on the PC surface would allow improving the detection of target biomolecules of low molecular weight (Figure 9).

## 5. Conclusions

In this study, several dextran-based types of quartz surface coatings, designed for the high-performance sorption of biomolecules and their subsequent detection by the PC SM biosensor, were successfully developed and studied. We have demonstrated that dextran with Mw 500 kDa and anchor epoxy groups had the maximal sorption capacity of biomolecules. The proposed approach would make it possible to enhance the sensitivity of the PC SM biosensor and also permit studying the binding process of low molecular weight molecules in real time.

## Figures and Tables

**Figure 1 polymers-14-00152-f001:**
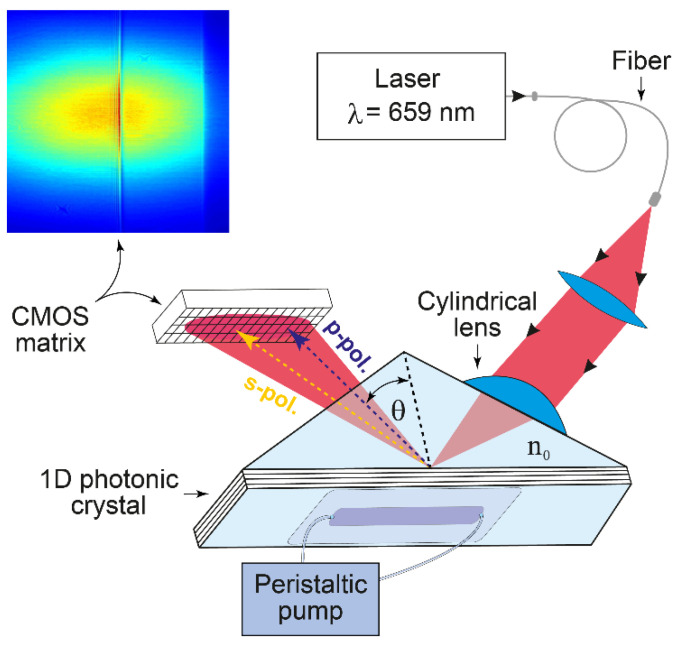
Schematic of the EVA 2.0 biosensor based on an angle interrogation of a PC SM. The beam of a stabilized laser is focused by a cylindrical lens in such a way that the excitation angle of one s-polarized PC surface mode (existing in this 1D PC) and total internal reflection angle (in p-polarization) are contained in the convergence angle of the beam. The reflected angular profile (see the color inset) is recorded by the complementary metal-oxide-semiconductor (CMOS) matrix.

**Figure 2 polymers-14-00152-f002:**
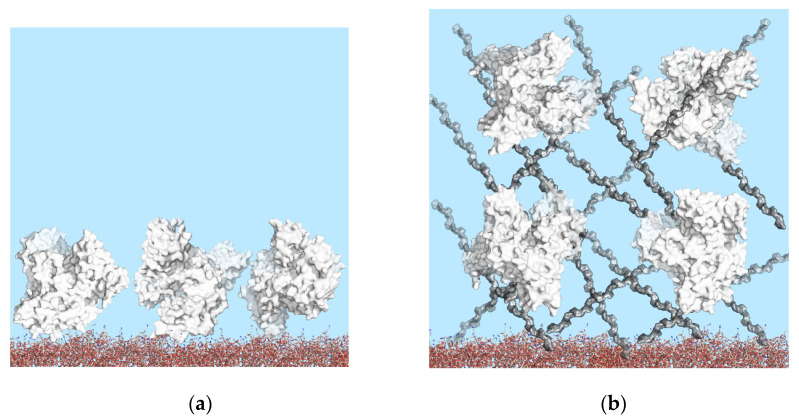
Schematic drawings of immobilized BSA on PC chip surfaces: (**a**) PC chip surface modified with APTES; (**b**) high surface-area-to-volume ratio 3D surface modified with dextrans. Role of surface geometry in binding site density.

**Figure 3 polymers-14-00152-f003:**
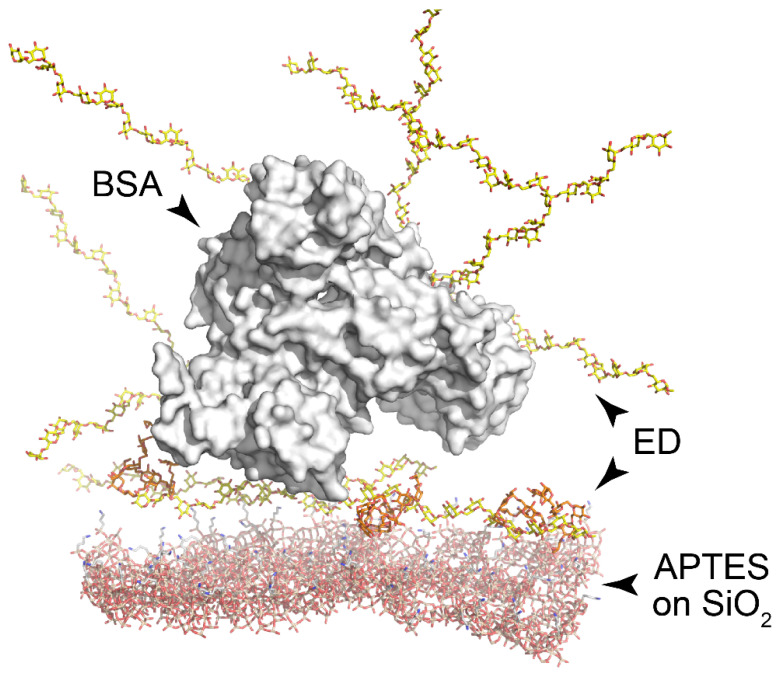
Visualization of macromolecular complexes at the PC surface. Silanized surface of silica (bottom, red); ED layer (yellow); BSA molecule (white surface). Amino groups are colored in blue.

**Figure 4 polymers-14-00152-f004:**
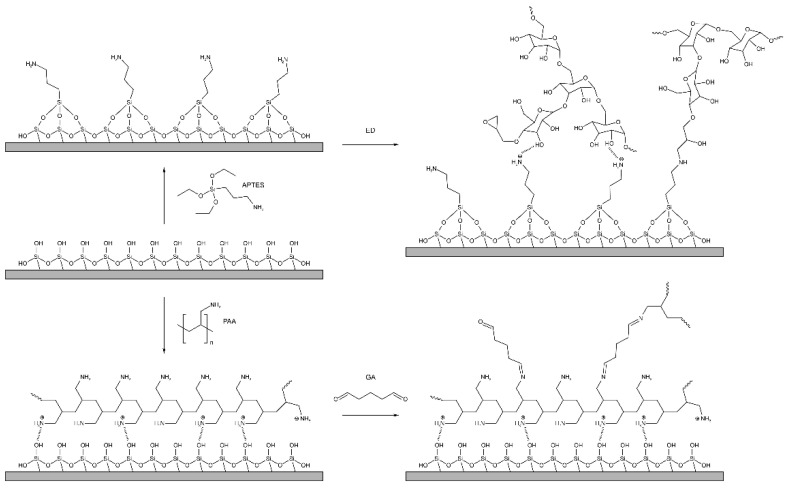
Modification of the chip surface: introduction of active amino groups on the chip surface (APTES treating) following ED modification for further dextran coupling and PAA and GA modification.

**Figure 5 polymers-14-00152-f005:**
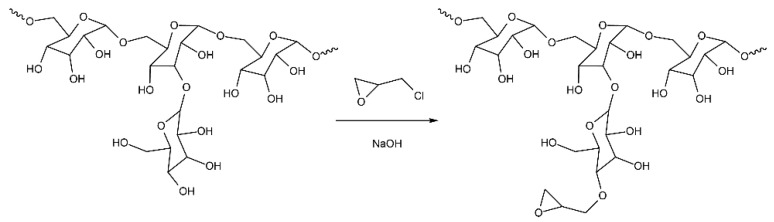
Scheme of ED preparation.

**Figure 6 polymers-14-00152-f006:**
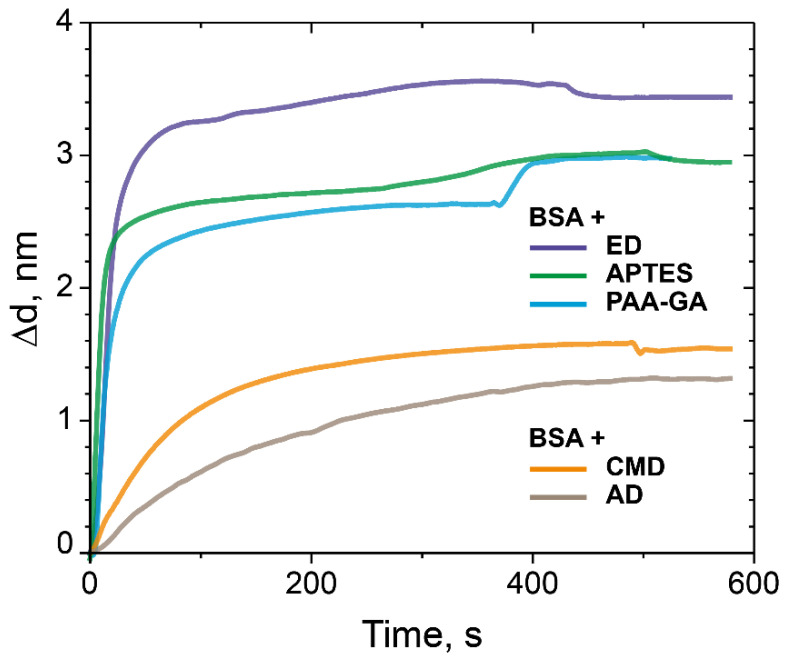
The change in the adlayer thickness upon the binding of BSA with the modified CMD, AD, ED-treated PC surface, and as controls—the PC surface treated with APTES and the PC surface modified with PAA and GA.

**Figure 7 polymers-14-00152-f007:**
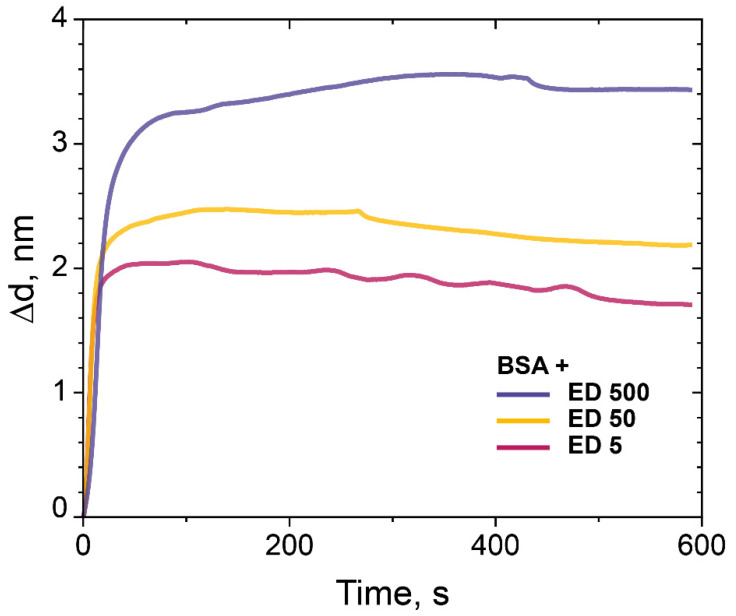
The change in the adlayer thickness upon binding of BSA (0.1 mg/mL) with the PC chip surface modified with ED (Mw 5 kDa, 50 kDa, and 500 kDa).

**Figure 8 polymers-14-00152-f008:**
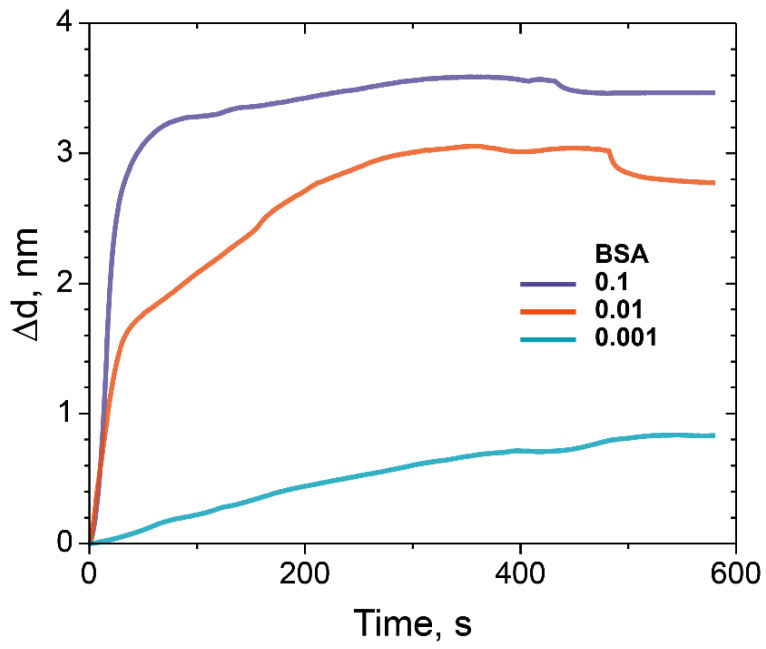
The change in the adlayer thickness upon the binding of BSA with different concentrations on ED-modified PC chip surface (Mw 500 kDa): 0.1 mg/mL (purple), 0.01 mg/mL (red), and 0.001 mg/mL (blue).

**Figure 9 polymers-14-00152-f009:**
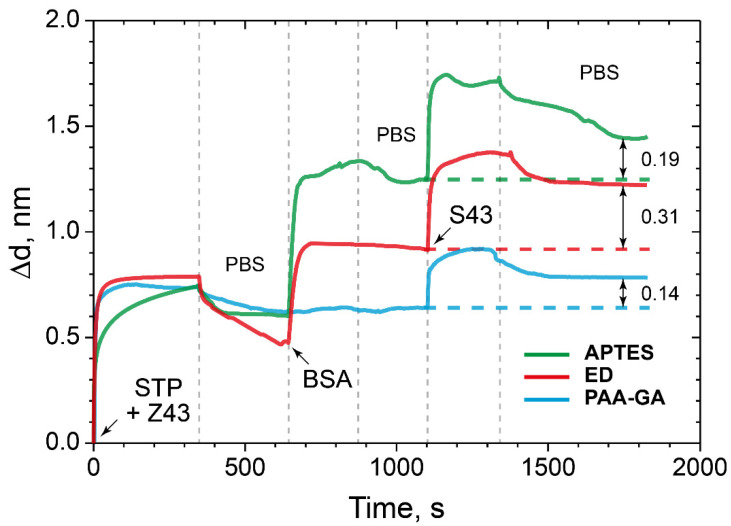
The change in the adlayer thickness upon binding of STP-biotinylated oligosensor with specific oligotarget: the PC surface modified with ED (red), APTES (green), and PAA and GA (blue).

**Figure 10 polymers-14-00152-f010:**
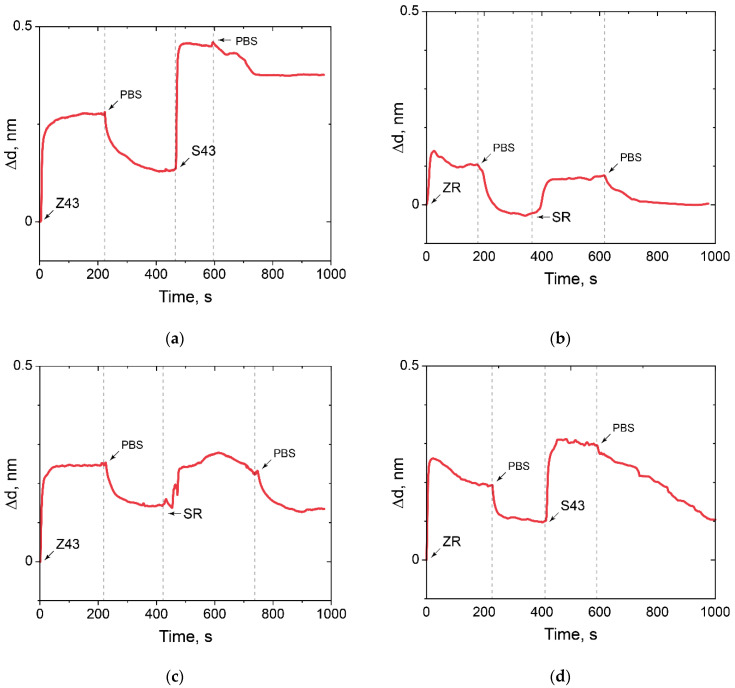
The change in the adlayer thickness upon positive and negative control of oligonucleotide hybridization. (**a**) Z43 is oligosensor, S43 is oligotarget; (**b**) ZR is oligosensor, SR is oligotarget; (**c**) ZR is oligosensor, S43 is oligotarget; (**d**) Z43 is oligosensor, SR is oligotarget.

**Figure 11 polymers-14-00152-f011:**
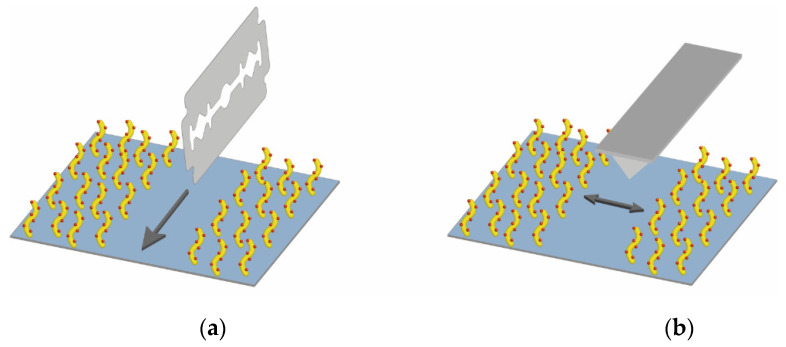
Schematic drawing of (**a**) sample preparation for AFM and (**b**) scanning process.

**Table 1 polymers-14-00152-t001:** Sequences of oligonucleotides used in the experiments.

Name	Sequence, 5′→3′	Length, nt	Mw, Da
S43, oligotarget	GTA TAC GTT GCT GCA CCT CCC GCA CCC GGT GCG ATT CTG CGT CCA GTT TCC GTC CCC TCT CG	62	18827.1
Z43, oligosensor	GGA GGT GCA GCA ACG TAT AC-(Biotin-TEG)	20	6761.1
SR, random oligotarget	TTC GAC ACT GCT AAA ATC ATT AAT CAA CCT GGA TAT TCT CTC GTG TTC TAT GCG TCT CTC AA	62	18902.4
ZR, random oligosensor	ACC GGT TAC CGC CTC CAC TG-(Biotin-TEG)	20	6583.9

**Table 2 polymers-14-00152-t002:** Adlayer thickness (Δd) upon the binding of the STP–biotinylated oligosensor with specific oligotargets: the PC surface modified with PAA and GA, APTES, and ED, and detection capacity increasing of the target low molecular weight molecules–oligonucleotide ED-modified PC surface in comparison with the control surfaces modified with APTES, PAA, and GA.

Δd^PAA−GA^, nm	Δd^APTES^, nm	Δd^ED^, nm	(Δd^ED^−Δd^PAA−GA^)/ Δd^PAA-GA^, %	(Δd^ED^−Δd^APTES^)/ Δd^APTES^, %
0.14	0.19	0.31	121	63

## Data Availability

Not applicable.

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
