# Peer review of "The Elaboration of Effective Coatings for Photonic Crystal Chips in Optical Biosensors"

_polymers, 2021, doi:10.3390/polym14010152_

Round 1
Reviewer 1 Report
The research article by S. Sizova et al. describes a detailed study on different kind of quartz surface coating for high-performance detection of proteins and biomolecules and their subsequent detection by PC-SM biosensor. In particular, the authors investigated the surface modification with dextran at different molecular weight and with different functionalities, namely epoxy (E), aldehyde (A) and carboxymethyl (CM) groups. The maximal sorption capacity of biomolecules has been demonstrated for dextran at 500KDa and with epoxy groups. In general, the manuscript is well-written and organized; the procedure described and in particular the coatings explored can be applied to enhance the biosensor sensitivity as well as to investigate low molecular weight binding process in real-time. However, in some points, i.e. in the introductive part, the debate could be broadened to increase the readability and references are missing or can be increased. Some of the results presented should be better described and also discussed (in particular, paragraphs 3.4 and 3.5)
Recommendations to improve the quality of the paper are listed below:
-at line 49 authors state "To date, various methods have been developed for preparing and recharging the surface of a biosensor substrate." For completeness, some references should be added here. There are several type of surface modification, including physical and chemical modification; a brief introduction can help the reading. A huge part of literature is dedicated to this issue; authors can critically use following review or the works cited therein (Sensors 2015, 15(1), 1635-1675; https://doi.org/10.3390/s150101635; Sensors and Actuators B: Chemical, 2018, 265, 161-173 https://doi.org/10.1016/j.snb.2018.02.190; Biosensors 2020, 10(11), 182; https://doi.org/10.3390/bios10110182)
-at lines 50- 52, authors state "the methods used in surface plasmon resonance (SPR) technique may, theoretically, also be suitable for modifying the surface of a photonic crystal (PC)" . Which methods are you referring to? Are those methods exclusively used in SPR techniques? Also in this case they should be mentioned. Please add them.
-Since the research deals with surface functionalization with organosilanes and subsequent coating with dextran having different functional groups, the case of two-step functionalization to control neuronal adhesion should be mentioned. (Langmuir 2016, 32, 25, 6319–6327 - https://doi.org/10.1021/acs.langmuir.6b01352 ). It is the clear demonstration of how chemical surface modification can be used both to detect isolated biomolecules (the case of sensors) and within a whole living cell (for different applications ranging, e.g. from electrophysiology to cell patterning)
-at line 76 authors state "There has been recent interest in polysaccharide coatings for biomaterials applications and for biosensors." Without references the sentence appears too generic; polysaccharides have been used both to enhance biointeraction as well as to increase the antiadhesive properties of biometarials. For completeness, a brief mention to this aspect help the reading and give the idea of the huge field. References should be added here. Authors can critically use following works (ACS Biomater. Sci. Eng. 2019, 5, 11, 5825–5832 https://doi.org/10.1021/acsbiomaterials.9b01288; Molecules 2021, 26(15), 4499; https://doi.org/10.3390/molecules26154499)
- line 117. APTES, PAA and GA in materials and methods section should be written in the extended form and not only as acronym.
- at line 256 authors state " The dextran activation process consists in the formation of free active groups within the dextran molecule. Another method of activation is to introduce active groups into the dextran molecule using a linker. These techniques are often combined " Please add references.
- Fig 4 clearly represent silicon functionalization with APTES; what about PAA and GA functionalization? May the authors describe as well those functionalization (not necessarily with another figure)?.
- Figure 6 clearly show the increase of ad thickness upon binding BSA using ED Mw 500kDa; How the authors justify the decrease when using CMD and AD? Please add comments.
-The Figure 9 caption is not coherent with profile colors. In general, the detection of low Mw biomolecules by PC SM biosensor can be better described.
-Similarly, Figure 10 caption is not coherent: change panels (c) and (d). In particular, in the cases of ZR as oligosensor (panels b and d) why Δd after 400 s is so different? In the first case it even goes below zero, whereas in the second is around 0.1. How many times the measurements have been repeated? There is an error bar? Please add comments.
-Figure 11; ok the schematic representation - even if to be precise the cantilever in panel (b) should be tilted 90° - but I would have expected AFM images. neither in SI there are, as Fig S1 is not AFM images but only profiles. In the present form is quite impossible evaluate the contribution from AFM measurements. Moreover, how did you manage images to obtain the topography profiles in Fig S1? Did you make any flatten? Please add comments otherwise the part of AFM measurements appears superfluous and can be removed.
Reviewer 2 Report
In this paper, the authors proposed and studied several types of quartz surface coatings, designed for high-performance sorption of biomolecules and their subsequent detection by a photonic crystal surface mode (PC SM) biosensor. Generally, the methods and idea are not novel, which can be found from previous reports, besides, the results and discussions are not clear to readers. Hence, it is hard for the reviewer to provide a more positive suggestion.
Major issues:
1, For the simulation, there is no details for anything, like the parameters, meshing process, etc.;
2, there is not sufficient characterization about the surface modification, like AFM or FTIR.
3, for the introduction, it is not clear for the reason of this design.
Reviewer 3 Report
Dear Editor and Authors,
I read carefully the article entitled ‘’The elaboration of effective coatings for photonic crystal chips in optical biosensors’’. I can conclude that is a nice preliminary study, but some issues must be clarified:
- Please add the method for estimating the number of aldehyde groups in AD in the Supplementary File.
- Please edit Figure 5.
- Please check the English.
Author Response
We thank the Reviewer for taking their time to read our manuscript and write a response.
Point 1: Please add the method for estimating the number of aldehyde groups in AD in the Supplementary File.
Response 1: The method of the aldehyde groups concentration measurement has been reported previously [1] and adapted for AD. We have changed the reference [34] (Lines 156 - 157) in the Manuscript and added the method in Supplementary File.
- Generalova, A.N., Sizova, S.V., Zdobnova, T.A., Zarifullina, M.M., Artemyev, M.V., Baranov, A.V., Oleinikov, V.A., Zubov, V.P., Deyev, S.M. Submicron polymer particles containing fluorescent semiconductor nanocrystals CdSe/ZnS for bioassays. Nanomedicine, 2011, 6(2), 195–209. doi:10.2217/nnm.10.162.
Point 2: Please edit Figure 5.
Response 2: Thank you, we agree with this comment. We have changed the Figure 5.
Point 3: Please check the English.
Response 3: We have checked the English.
Round 2
Reviewer 1 Report
The authors addressed all the point raised up in my previous revision round.
Reviewer 2 Report
Many thanks for the authors' responses and the SI information. However, it is still hard for the reviewer to provide a positive recommendation, mainly because the following reasons:
1, the design of this paper is not novel, as agreed by the authors: there is no obvious contribution to this field, based on the current data and results;
2, the key point of this work lies in the capture of BSA by dextran matrix, however, the mechanism is not well and clearly explained.
3, the selectivity of this paper has not been fully demonstrated, BSA molecules could be captured, while, other proteins also work.
4, the design of this work is not significant enough to be published in polymers.